# Prognostic Value of Incomplete Revascularization after Percutaneous Coronary Intervention Following Acute Coronary Syndrome: Focus on CKD Patients

**DOI:** 10.3390/jcm8060810

**Published:** 2019-06-06

**Authors:** Thomas Cardi, Anas Kayali, Antonin Trimaille, Benjamin Marchandot, Jessica Ristorto, Viet Anh Hoang, Sébastien Hess, Marion Kibler, Laurence Jesel, Patrick Ohlmann, Olivier Morel

**Affiliations:** 1Pôle d’Activité Médico-Chirurgicale Cardio-Vasculaire, Nouvel Hôpital Civil, Centre Hospitalier Universitaire, Université de Strasbourg, 67090 Strasbourg, France; thomas.cardi@chru-strasbourg.fr (T.C.); anas.kayali@chru-strasbourg.fr (A.K.); benjamin.marchandot@chru-strasbourg.fr (B.M.); jessica.ristorto@chru-strasbourg.fr (J.R.); vietanhhoang78@yahoo.fr (V.A.H.); sebastien.hess@chru-strasbourg.fr (S.H.); mario.kibler@chru-strasbourg.fr (M.K.); laurence.jesel@chru-strasbourg.fr (L.J.); patrick.ohlmann@chru-strasbourg.fr (P.O.); olivier.morel@chru-strasbourg.fr (O.M.); 2Vietnam National Heart Institute, Bach Mai Hospital, 78 Giai Phong, Dong Da, 10000 Hanoi, Vietnam; 3Regenerative Nanomedicine, UMR 1260, INSERM (French National Institute of Health and Medical Research), FMTS (Fédération de Médecine Translationnelle de l’Université de Strasbourg), Université de Strasbourg, Faculté de Médecine, 11 rue Humann, 67085 Strasbourg, France

**Keywords:** chronic kidney disease, SYNTAX score, residual SYNTAX score, incomplete revascularization, coronary artery disease

## Abstract

Background: Residual coronary artery disease (CAD) has been associated with worsened prognosis in patients undergoing percutaneous coronary intervention (PCI) for acute coronary syndromes (ACS). The residual SYNTAX Score (rSS) aims to assess residual CAD after PCI. The association between kidney function and rSS has not been investigated in ACS patients. In this study, we sought to determine whether chronic kidney disease (CKD) patients exhibit more incomplete revascularization following stage revascularization procedures by PCI. We evaluated the impact of incomplete revascularization on the occurrence of major cardiovascular events (MACE) at one-year follow-up. Methods: A total of 831 ACS patients undergoing PCI were divided into 3 subgroups according to their estimated Glomerular Filtration Rate (eGFR): 695 with eGFR ≥ 60 mL/min/1.73 m², 108 with eGFR 60–30 mL/min/1.73 m², 28 with eGFR < 30 mL/min/1.73 m². Initial SYNTAX score (SS) and rSS were calculated for all patients. Incomplete revascularization was defined by rSS > 8. The primary endpoint was the occurrence of MACE (all-cause mortality, myocardial infarction (MI), repeated revascularization except from planned revascularization, stroke and definite or probable recurrent stent thrombosis) one year after the index procedure. Results: Severe CKD patients had significantly higher MACE (12.0% vs. 25.9% vs. 35.7%; *p* < 0.001), all-cause mortality, cardiovascular mortality and heart failure events. Patients with rSS > 8 had higher MACE, all-cause and cardiovascular mortality. CKD was an independent predictive factor of rSS > 8 (HR: 1.65, 95% CI: 1.01 to 2.71; *p* = 0.048). Multivariate analysis identified rSS > 8, but not CKD, as an independent predictor of cardiac death and MACE. Conclusion: In ACS, CKD is predictive of incomplete revascularization, which stands out as a strong predictor of adverse cardiovascular outcomes including cardiac death and MACE.

## 1. Introduction

With respect to the importance of non-culprit lesions in acute coronary syndromes (ACS), the current characterization of residual atherosclerotic burden by the generic denomination “multiple vessel disease” appears insufficient to accurately assess the risk of adverse cardiovascular outcomes [1,2,3]. Focused attention on concomitant and additional coronary lesions is crucial to guide clinical decisions in the field of coronary artery disease (CAD) and improve mid- and long-term prognostics following percutaneous coronary intervention (PCI). Therefore, the residual SYNTAX Score (rSS) has emerged as an angiographic scoring system aimed to assess residual (CAD) after (PCI) [4]. A recent study has established that a staged PCI that achieves reasonably complete revascularization (rSS ≤ 8) improves mid-term survival and reduces the incidence of repeated PCI in patients with ST-segment elevation myocardial infarctions (STEMI) and multivessel disease [5]. By contrast, significant residual CAD assessed by higher rSS, confers a worsened prognosis in patients undergoing PCI. Diabetes, previous myocardial infarction (MI) and multivessel coronary artery disease are currently established as independent predictors of high rSS [1,4]. Up to now, the association between kidney function and rSS has not been adequately investigated in ACS patients. In the present prospective study, we sought to determine the incidence of incomplete revascularization by calculating rSS among chronic kidney disease (CKD) patients with staged revascularization procedures by PCI in ACS. Moreover, we sought to investigate the impact of incomplete revascularization on the occurrence of major adverse cardiovascular events (MACE) at one-year follow-up.

## 2. Methods

### 2.1. Patients

This study enrolled all consecutive ACS patients undergoing PCI at our institution (Nouvel Hôpital Civil, Université de Strasbourg, Strasbourg, France) between February 2014 and February 2016. The study was performed in accordance with the Declaration of Helsinki. The protocol was approved by the ethics committee, and informed written consent was obtained from all patients. The flow-chart of the study is shown in Figure 1.

### 2.2. Chronic Kidney Disease Staging

Baseline serum creatinine levels were assessed at admission in all patients. The estimated glomerular filtration rate (eGFR) was calculated using the MDRD (Modification of Diet in Renal Disease) formula [6]. Patients were divided into 3 subgroups according to their eGFR levels: ≥60 mL/min/1.73 m^2^; ≥30 and <60 mL/min/1.73 m^2^; and <30 mL/min/1.73 m^2^. Patient’s baseline characteristics included age, gender, body mass index (BMI), history of dyslipidemia, smoking status at admission, history of hypertension (HT), history of diabetes mellitus (DM), personal and family history of cardiovascular disease, left ventricle ejection fraction (LVEF) and discharge medication. Patients with prior coronary artery bypass grafting (CABG) were excluded from our analysis. 

### 2.3. Collection of Data

Follow-up information was obtained using a written questionnaire via a telephone interview with the cardiologist, referring physician or patient. In the absence of response, the patient’s electronic medical file was consulted. 

### 2.4. Calculation of SYNTAX Score (SS), Residual SYNTAX Score (rSS) and SYNTAX Index Revascularization (SRI)

The SYNTAX Score (SS) was calculated from the pre-procedural angiogram, in which each coronary lesion producing >50% diameter stenosis in vessels >1.5 mm by visual estimation was scored separately using the SS algorithm, and added to obtain the overall SS. Staged PCI was performed in patients with angiographic stenosis ≥70% or who demonstrated residual ischemia assessed either by fractional flow reserve (FFR) or by perfusion myocardial tomography. Staged PCI was performed within 30 days following the index ACS. The residual SYNTAX scores (rSS) was defined as the SS recalculated after staged PCI and was calculated in all patients enrolled in this study. To calculate the rSS, the final post-PCI angiogram was scored to assess untreated disease after staged PCI. A dedicated interventional cardiologist, who was blinded to both baseline characteristics and clinical outcomes reviewed all post-procedural angiograms. Similarly, the SYNTAX revascularization index (SRI) which stands as an angiographic index tool aimed to quantify the proportion of revascularized myocardium was calculated and defined as: 100 (1-rSS/baseline SS) (%). 

### 2.5. Study Endpoints

The primary endpoint was the major adverse cardiac events (MACE) rate, which was defined as the composite of all-cause mortality, myocardial infarction (MI), repeat revascularization except from planned revascularization, stroke and probable or definite recurrent stent thrombosis at one-year follow-up.

The secondary end points were all-cause mortality, cardiovascular mortality (defined as any death with demonstrable cardiovascular cause or any death that was not clearly attributable to a non-cardiovascular cause), ST-segment elevation myocardial infarction (STEMI) defined as a new ST-segment elevation in two consecutive leads associated with increased biochemical myocardial necrosis markers, non–ST-segment elevation myocardial infarction (NSTEMI) defined as the occurrence of ischemic symptoms, ST-segment depression and/or T-wave abnormalities and an increase of biochemical myocardial necrosis markers, stent thrombosis (according to the definition of the Academic Research Consortium), angina, hemorrhagic events (according to Bleeding Academic Research Consortium definition), and rehospitalization for heart failure. 

The extent of P2Y12 inhibition by thienopyridines or ticagrelor was established using the vasodilator stimulated phosphoprotein (VASP) assay as previously described [7,8].

### 2.6. Statistical Analysis

Continuous variables are expressed as mean +/− standard deviation (SD); categorical variables are expressed as frequencies and percentages (%). Continuous variables between 2 groups were compared by the Student t test or by the Mann-Whitney test as appropriate. Fisher’s exact test was used for comparison of categorical variables. Continuous variables were analyzed for normal distribution using the Shapiro-Wilk test. Time to event was defined as the time from PCI to the date of event, with patients censored at death, loss to follow-up, or end of the study. Correlations were calculated using the Spearman test. 

Kaplan-Meier analyses were used to construct survival plots of time to death after PCI and compared using the log-rank test. 

Multivariate analysis of survival rates was done using Cox models. Variables with *p* < 0.05 in univariate analysis were entered into a stepwise ascending multivariate analysis. In addition, variables linked to residual SS were not taken into account in the multivariate analysis for MACE and cardiac death (i.e.， three-vessel disease and SS). 

The results of the Cox regression are presented as hazard ratios (HR), their 95% confidence intervals (CIs), and p values. A *p* value < 0.05 was considered statistically significant. Statistical analysis was performed using SPSS version 13.0 software (SPSS Inc., Chicago, IL, USA).

## 3. Results

### 3.1. Patient’s Baseline Characteristics

A total of 831 consecutive ACS patients treated with PCI were enrolled in this study. Patients were divided into 3 groups according to their baseline eGFR level: 695 patients with GFR ≥ 60 mL/min/1.73 m² were referred to as the “no CKD” group (group 1), 108 patients with GFR 59–30 mL/min/1.73 m² as the “moderate CKD” group (group 2) and finally, 28 patients with GFR < 30 mL/min/1.73 m² as the “severe CKD” group (group 3). Baseline demographic, clinical, biological and angiographic characteristics are described in Table 1, Table 2 and Table 3. As previously published by several groups including ours, the extent of P2Y12 inhibition as assessed by platelet reactivity index (PRI) VASP was inversely related to renal function [8]. 

Compared to the “no CKD” group, patients with advanced CKD were older, showed increased prevalence of cardiovascular risk factors including diabetes mellitus, hypertension, dyslipidemia and displayed lower LVEF and more frequent right ventricular dysfunction. As expected, a history of cardiovascular disease was more frequently recorded in CKD patients. Likewise, higher burden of CAD could be demonstrated at baseline in CKD patients with higher SS at baseline and increased incidence of three-vessel coronary artery disease, as reported previously [9,10]. Staged procedures were achieved similarly in 24.9% of patients with GFR ≥ 60 mL/min/1.73 m², 29% of patients with GFR 30–59 mL/min/1.73 m² and 21.4% of patients with GFR < 30 mL/min/1.73 m² (*p* = 0.593). As expected, a significant relationship between baseline SS and rSS could be established (*r* = 0.535; *p* < 0.001). The mean SS was 13 ± 8 in patients with one- or two-vessel disease and 25 ± 10 in three-vessel disease (*p* < 0.001). Likewise, rSS was 2.9 ± 4.8 in patients with one or two-vessel disease and 8.6 ± 8.3 in patients with three-vessel disease (*p* < 0.001). Following staged procedures, higher rSS and lower SRI could be evidenced as eGFR declined (Figure 2). The proportions of rSS > 8 among the 3 groups were, respectively, 16.3%, 33.3% and 46.4% (*p* < 0.001). A 2- to 3-fold elevation of incomplete revascularization rate was evidenced as kidney function declined (Table 3). eGFR was inversely related to baseline SS (*r* = −0.163; *p* < 0.001) and rSS (*r* = −0.172; *p* < 0.001).

### 3.2. Impact of CKD on Cardiovascular Outcomes

Clinical outcomes were available in 826 patients (99.39%) with a mean follow-up of 324 days. Five patients were lost to follow-up. Major adverse cardiac events (MACE) at 1 year occurred in 121 patients (14.5%).

The rates of all-cause death, cardiovascular mortality, rehospitalization for heart failure and MACE were significantly higher among patients with CKD compared to those with preserved eGFR (Figure 3, Table 4).

### 3.3. Impact of Incomplete Revascularization (rSS > 8) on Cardiovascular Outcomes

Compared to the patients with optimal revascularization, patients with rSS > 8 had significantly higher risk of all cause and cardiovascular mortality, STEMI, CABG, stroke, angina, heart failure and MACE (Figure 4, Table 5). 

### 3.4. Predictors of rSS > 8 (Incomplete Revascularization)

By univariate Cox regression analysis, age, diabetes mellitus, hypertension, dyslipemia, atrial fibrillation, peripheral artery disease, prior NSTEMI, CKD, three-vessel disease were significant predictors of a rSS > 8 (Table 6). Multivariate Cox regression analysis identified age > 75 years, diabetes mellitus, CKD and three-vessel disease as independent predictors of a rSS > 8 (Table 6).

### 3.5. Predictors of Cardiac Death and MACE

By univariate Cox analysis, Killip class ≥ II, CKD, atrial fibrillation, peripheral artery disease, LVEF ≤ 40%, three-vessel disease, SS > 22, rSS > 8 and SRI < 70% were significant predictors of cardiac death at one year follow-up (Table 7). The occurrence of MACE was directly related to NSTEMI, hypertension, diabetes mellitus, CKD, atrial fibrillation, peripheral artery disease, Killip class ≥ II, LVEF ≤ 40%, three-vessel disease, SS > 22 and rSS > 8. By contrast, STEMI, the use of ACE/ARB, beta blocker or statin and one-vessel disease were associated with a lower risk of MACE (Table 7). Multivariate Cox regression analysis identified a rSS > 8, but not diabetes mellitus or CKD, as an independent predictor of cardiac death and MACE (Table 8). 

## 4. Discussion

The current report drawn from a cohort of 831 consecutive ACS patients, is the first study to quantify the extent and severity of coronary artery disease prior and after staged revascularization by PCI according to the extent of kidney disease. The main finding of this study is that CKD is predictive of incomplete revascularization during ACS as assessed by higher rSS values. The importance of rSS on adverse outcomes is emphasized by the evidence that incomplete revascularization, but not CKD or diabetes mellitus, is a strong independent predictor of one-year mortality, cardiac mortality and MACE following PCI. 

### 4.1. Impact of CKD on Adverse Cardiovascular Outcomes Following PCI

Patients with CKD represent an increasing proportion of the population undergoing PCI and up to 40% of the patients treated for ACS [11]. Several studies and registries in the context of ACS and elective PCI have reported a negative association between CKD and mortality, stent thrombosis, post procedural ischemic events on the one hand, and bleeding events on the other hand [11,12,13,14,15]. The mechanisms through which CKD affects the clinical outcome are believed to be multifactorial and may include increased cardiovascular comorbidities such as diabetes mellitus, enhanced oxidative stress, endothelial dysfunction, persistent micro-inflammation, coronary calcification, higher burden of yellow atherosclerotic plaques and platelet activation [16,17]. Other factors contributing to adverse outcomes include insufficient use of well-proven therapies, lower inhibition by thienopyridines [7,18] and reluctance to perform coronary angiography, which is mostly motivated by advanced age, the presence of multiple comorbidities, bleeding risk and an enhanced risk of contrast induced nephropathy as eGFR declines. A first insight into the deleterious role of undergoing revascularization in CKD came from Chertow and coworkers. They demonstrated that inappropriately low rates of coronary angiography in elderly individuals with renal insufficiency indicated by itself, an enhanced risk of adverse outcomes [19]. Likewise, another study has suggested in patients with multivessel CAD that complete revascularization by drug-eluting stents (DES) reduced cardiac death and myocardial infarction incidence. The observed benefit of complete revascularization was maximal in diabetes, low ejection fraction and low eGFR patients [20]. Challenging this view, a recent retrospective multivariate analysis has shown that incomplete revascularization was not independently predictive of adverse outcomes in octogenarians and supports a more conservative approach [21]. In the present study, increased baseline coronary artery disease burden was evidenced as kidney function declined and witnessed by higher prevalence of three-vessel disease, baseline SS, % of patients with SS > 22 or 32 (Table 3). Baseline SS calculated in the whole cohort (16.4 ± 10) was higher than the one proposed in the original report by Genereux (12.8 ± 6) and derived from a post hoc analysis of the ACUITY (Acute Catheterization and Urgent Intervention Triage Strategy) trial [4]. In our work, staged PCI was equally performed among the 3 groups but could not totally blunt baseline differences, rSS, and SRI < 70%, which happened to be at their highest levels in patients with severe CKD. The proportion of patients with rSS > 8 observed in this real-world study (19.5%) is in line with the original report by Genereux (18.7%) [4] and a recent report by Khan in STEMI patients (21.8%) [1]. The question whether CKD, or incomplete revascularization associated with CKD independently hamper cardiovascular outcomes has not been addressed up to now. In our study, CKD along with history of diabetes mellitus, age > 75 years and three-vessel disease were demonstrated to be independent predictors of rSS > 8. In an attempt to delineate the respective contribution of CKD, diabetes mellitus and incomplete revascularization on cardiac mortality and MACE following ACS, multivariate analysis was performed. The present data suggest that incomplete revascularization plays a major role in the determination of adverse outcomes; and both CKD and diabetes mellitus are no longer independently associated with cardiac mortality/MACE when rSS was taken into account. The most commonly accepted reasons for incomplete revascularization in CKD patients are generally multifactorial and may include the following: (i) ignorance of NSTE-ACS risk (despite paradoxically high calculated GRACE-risk scores), (ii) ignorance of potential treatment benefit in these patients, (iii) fear of an immediate complication such as contrast induced nephropathy or bleeding, (iv) co-morbidities, and (v) absence of population-specific clinical trial data. 

### 4.2. Impact of Incomplete Revascularization on Adverse Cardiovascular Outcomes Following PCI

The present study extends previous reports that have demonstrated a strong relationship between incomplete revascularization and worse cardiovascular outcomes in several circumstances including ACS [1,4,22,23]. However, a marked variability in the prognostic value of incomplete revascularization has been reported and possibly reflects the lack of consensus on the definition of optimal revascularization [20]. Other important drawbacks of these studies, including the present one, rely on the definition of incomplete revascularization based on percent diameter stenosis derived from visual estimation. As nicely depicted by the Fractional Flow Reserve Versus Angiography for Multivessel Disease Evaluation (FAME) study, nearly one third of angiographically significant stenosis were actually hemodynamically not significant [24]. Therefore, the poor correlation between the angiogram and the presence of vessel related ischemia could cloud the relationship between the degree of revascularization and clinical outcome [25]. In line with this view, recent data derived from the FAME study have demonstrated that residual angiographic lesions that are not functionally significant (FFR > 0.80) do not reflect residual ischemia or predict adverse outcomes [26]. Despite these limitations, the validation of incomplete revascularization in clinical practice assessed by a rSS > 8 was originally addressed by Genereux and coworkers as an important marker of adverse outcomes in ACS patients. 

The rSS aims to represent a reliable measurement of the myocardial ischemia burden. It depends on the location of the coronary disease, the importance in supplying blood to the jeopardized myocardium, and the anatomic complexity associated with the obstructive disease [23]. 

In the work by Genereux, age, insulin-treated diabetes, hypertension, smoking, elevated biomarkers or ST-segment deviation and lower ejection fraction were more frequent in patients with incomplete revascularization [4]. Although the demonstration that incomplete revascularization is associated with a poorer prognosis does not necessarily imply that revascularization of residual stenosis visible on the angiogram improves prognosis, these data might indicate that the tracking of residual ischemia is highly warranted in high risk patients such as those with CKD. In that setting, given the high prevalence of incomplete revascularization observed in CKD patients (33.3% and 46.4%) following staged PCI, alternative strategies of revascularization such as CABG or hybrid PCI plus CABG should be evaluated by the heart team unless the interventional cardiologist is confident that the level of incomplete revascularization will be low. Indeed, a recent large meta-analysis has emphasized that complete revascularization is more commonly achieved with CABG [27]. 

### 4.3. Study Limitations

This study includes several limitations. (i) As the SYNTAX score is not designed to assess angiographic complexity in patients with CABG, those patients were excluded from our analysis. (ii) The SYNTAX score was assessed by only one investigator (A.K.). Intra-individual variability assessed by a new evaluation of 10 random angiograms was less than 5%. (iii) The small number of patients with severe CKD limited the interpretation of our data. (iv) The functional assessment of angiographic stenosis by FFR was not systematically investigated. (v) The SYNTAX score is basically aimed to assess the technical complexity of percutaneous coronary interventions. It is therefore conceivable that some of the features used to calculate the score have a marked impact on the success of the procedure (e.g., tortuosity of calcification) but may not be added as prognostic information for future atherosclerosis-related events. (vi) The cardiovascular events were not adjudicated by an independent committee. Finally, as with similar evaluations of registry data, there are inherent limitations for this type of study, mainly related to known or unknown confounding factors. The results of this report should therefore be considered as hypothesis generating and prospective trials are required to further evaluate the importance of achieving complete revascularization in CKD patients. 

## 5. Conclusions

In acute coronary syndrome, CKD is predictive of incomplete revascularization following staged PCI procedures. The importance of rSS on adverse outcomes is emphasized by our demonstration that incomplete revascularization, but not CKD nor diabetes mellitus, is a strong independent predictor of one-year mortality, cardiac mortality and MACE following PCI.

## Figures and Tables

**Figure 1 jcm-08-00810-f001:**
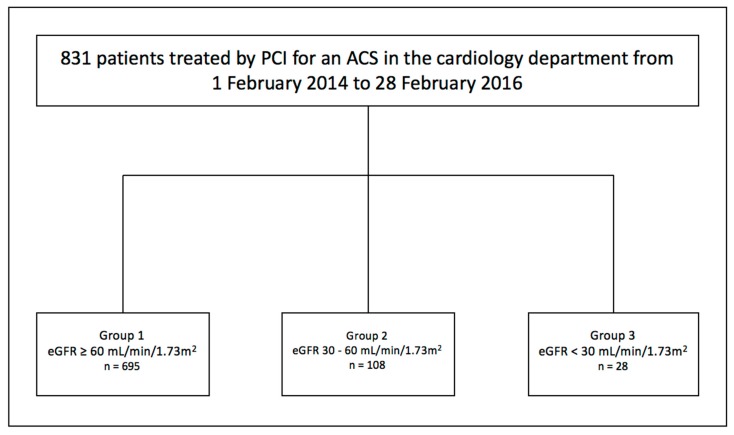
Flow-chart. *n*: number, ACS: Acute coronary syndrome, eGFR: estimated Glomerular filtration rate, PCI: Percutaneous coronary interventions.

**Figure 2 jcm-08-00810-f002:**
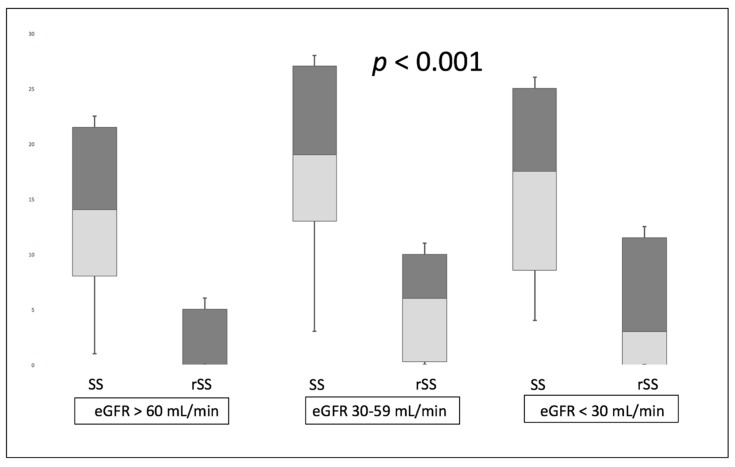
SYNTAX score and rSS according to renal function. SS: SYNTAX score; rSS: residual SYNTAX score.

**Figure 3 jcm-08-00810-f003:**
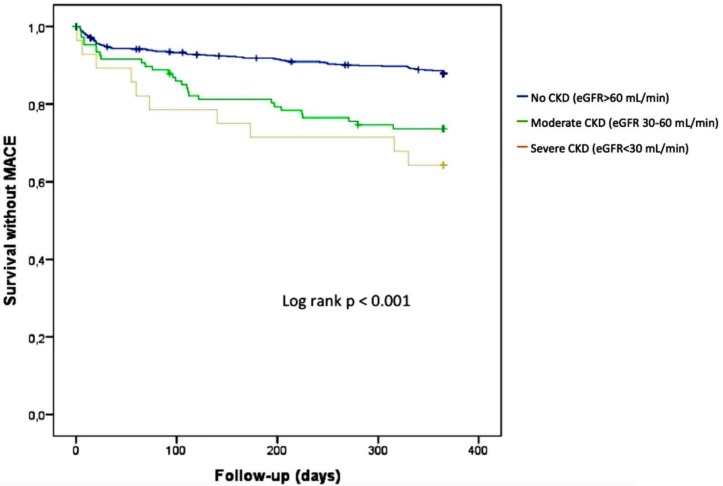
Kaplan Meier analysis for the probability of MACE event-free survival according to renal function.

**Figure 4 jcm-08-00810-f004:**
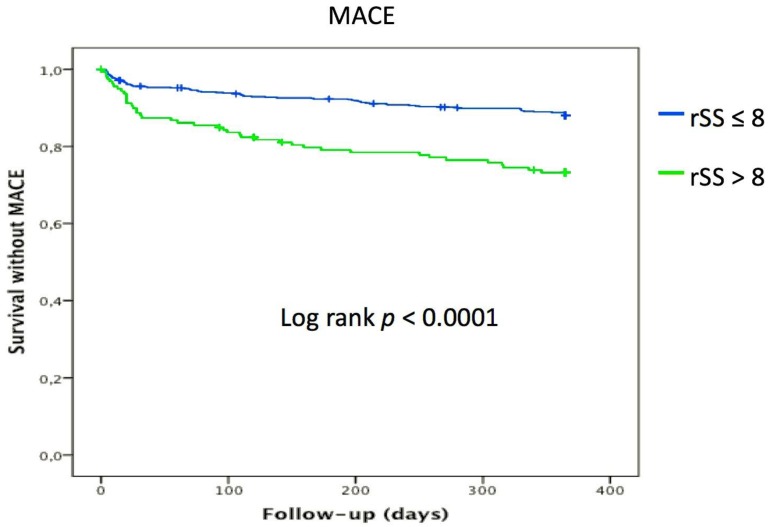
Kaplan Meier analysis for the probability of MACE event-free survival according to rSS.

**Table 1 jcm-08-00810-t001:** Baseline demographic and clinical characteristics according to renal function.

Variables	eGFR ≥ 60 *n* = 695	eGFR 30-59 *n* = 108	eGFR < 30 *n* = 28	*p* Value
Age	61.88 (±13.15)	76.79 (±12.49)	76.43 (±12.55)	<0.001
Gender (female)	153 (22%)	38 (35.2%)	10 (35.7%)	0.004
STEMI	392 (56.4%)	50 (46.3%)	15 (53.6%)	0.161
NSTEMI	268 (38.6%)	52 (48.1%)	12 (42.9%)	0.159
Unstable angina	35 (5%)	6 (5.6%)	1 (3.6%)	0.911
Diabetes mellitus	144 (20.7%)	44 (40.7%)	12 (42.9%)	<0.001
Hypertension	382 (55%)	89 (82.4%)	23 (82.1%)	<0.001
Current smoking	299 (43%)	19 (17.6%)	4 (14.3%)	<0.001
Dyslipidemia	314 (45.2%)	66 (61.1%)	20 (71.4%)	<0.001
Family history of CAD	160 (23%)	8 (7.4%)	4 (14.3%)	0.001
Prior STEMI	68 (9.80%)	14 (13%)	7 (25%)	0.028
Prior NSTEMI	44 (6.3%)	17 (15.7%)	6 (21.4%)	<0.001
Prior angioplasty	102 (14.7%)	24 (22.2%)	10 (35.7%)	0.003
Prior Stroke	35 (5%)	12 (11.1%)	6 (21.4%)	<0.001
Atrial fibrillation	40 (5.8%)	17 (15.9%)	7 (25%)	<0.001
PAD	41 (5.90%)	17 (15.9%)	6 (21.4%)	<0.001
Obesity	188 (27.1%)	19 (17.6%)	5 (17.9%)	0.237
Killip ≥ 2	105 (15.1%)	44 (40.7%)	14 (50%)	<0.001
LVEF (%)	52.51 (±11.44)	49.22 (±12.87)	45.25 (±12.21)	<0.001
LVEF ≤ 40%	125 (18%)	32 (29.6%)	12 (42.9%)	<0.001
RV dysfunction	42 (6%)	11 (10.2%)	9 (32.1%)	<0.001
Aspirin	688 (99%)	104 (96.3%)	27 (96.4%)	0.058
Clopidogrel	248 (35.7%)	74 (68.5%)	25 (89.3%)	<0.001
Ticagrelor	188 (27.1%)	18 (16.7%)	1 (3.6%)	<0.001
Prasugrel	249 (35.80%)	12 (11.1%)	0 (0%)	<0.001
ACE-I	571 (82.2%)	67 (62%)	14 (50%)	<0.001
ARB	57 (8.2%)	20 (18.5%)	4 (14.3%)	<0.001
Statin	671 (96.5%)	97 (89.8%)	24 (85.7%)	0.002
Betablocker	645 (92.9%)	90 (83.3%)	23 (82.1%)	0.001
VKA	67 (9.6%)	20 (18.5%)	9 (32.1%)	0.003
In hospital hemorrhage	17 (2.4%)	7 (6.5%)	3 (10.7%)	0.007
Acute kidney injury	54 (7.8%)	41 (38%)	14 (50%)	<0.001

Data are presented as mean ± SD or *n* (%). eGFR is expressed in mL/min/1.73 m^2^. eGFR: estimated Glomerular Filtration rate, STEMI: ST elevation myocardial infarction, NSTEMI: non STEMI, PAD: peripheral artery disease, LVEF: Left ventricle ejection fraction, ACE-I: Angiotensin converting enzyme - inhibitors, ARB: Angiotensin II receptor blockers, RV: Right Ventricular, VKA: Vitamin K antagonists.

**Table 2 jcm-08-00810-t002:** Biological parameters according to renal function.

Variables	eGFR ≥ 60 *n* = 695	eGFR 30-59 *n* = 108	eGFR < 30 *n* = 28	*p* Value
Creatinine (µmol/L)	71.31 ± 14.82	123.61 ± 25.76	262.29 ± 169.98	<0.001
eGFR (mL/min/1.73 m²)	85.77 ± 7.75	48.48 ± 8.13	21.93 ± 8.44	<0.001
Troponin at admission (µg/L)	10.01 ± 34.72	13.75 ± 36.54	15.56 ± 26.44	0.436
Troponin peak (µg/L)	70.93 ± 130.26	67.14 ± 123.98	75.24 ± 121.28	0.942
BNP (µg/L)	139.87 ± 253.3	462.74 ± 640.46	639.07 ± 599.53	<0.001
CRP (mg/L)	11.69 ± 27.21	34.72 ± 57.78	72.44 ± 92.99	<0.001
Total-cholesterol (mg/dL)	1.76 ± 0.58	1.56 ± 0.6	1.35 ± 0.69	<0.001
LDL-C (mg/dL)	1.09 ± 0.44	0.92 ± 0.45	0.76 ± 0.45	<0.001
HDL-C (mg/dL)	0.39 ± 0.15	0.4 ± 0.17	0.33 ± 0.16	0.065
Triglycerid (mg/dL)	1.37 ± 0.9	1.14 ± 0.66	1.14 ± 0.93	0.019
Glycated Hemoglobin (%)	5.55 ± 1.9	5.58 ± 2.16	5.57 ± 2.09	0.994
Hemoglobin (g/dL)	14.23 ± 1.69	13.15 ± 2.07	11.32 ± 2.16	<0.001
White blood cell count (×10^9^/L)	10.8 ± 4.07	11.22 ± 5.87	11.92 ± 5.98	0.323
Platelets (×10^9^/L)	242.5 ± 67.17	225.36 ± 90.02	222.64 ± 72.34	0.028
VASP (%)	23.61 ± 25.53	28.47 ± 29.19	48.44 ± 33.04	<0.001

Data are expressed as mean ± SD. eGFR is expressed in mL/min/1.73 m^2^. eGFR: estimated Glomerular Filtration rate, BNP: Brain natriuretic peptide, CRP: C-reactive protein, LDL-C: Low-Density Lipoprotein Cholesterol, HDL-C: High-Density Lipoprotein-Cholesterol, VASP: vasodilator stimulated phosphoprotein.

**Table 3 jcm-08-00810-t003:** Angiographic and PCI procedure characteristics according to renal function.

Variables	eGFR ≥ 60 *n* = 695	eGFR 30-59 *n* = 108	eGFR < 30 *n* = 28	*p* Value
One-vessel disease	340 (48.9%)	35 (32.4%)	11 (39.3%)	0.004
Three-vessel disease	144 (20.7%)	31 (28.7%)	10 (35.7%)	0.039
LAD	426 (61.3%)	75 (69.4%)	16 (57.1%)	0.228
Left main	28 (4%)	9 (8.3%)	2 (7.1%)	0.119
Bifurcation	28 (4%)	6 (5.6%)	1 (3.6%)	0.752
SYNTAX score (SS)	15.77 (±9.95)	19.27 (±10)	21.34 (±13.91)	<0.001
SS > 22	159 (22.8%)	37 (34.3%)	11 (39.3%)	0.007
SS > 32	53 (7.6%)	9 (8.3%)	7 (25%)	0.005
Stent implantation	655 (94.2%)	105 (97.2%)	28 (100%)	0.195
Total stent lenghts (mm)	29.99 (±20.35)	32.86 (±21.29)	34.39 (±19.51)	0.238
Maximum Stent Diameter per lesion (mm)	2.91 (±0.82)	2.95 (±0.65)	3.18 (±0.39)	0,199
Thrombectomy	102 (14.7%)	10 (9.3%)	4 (14.3%)	0.319
Irradiation time (min)	11.42 (±8.34)	12.36 (±10.98)	12.38 (±7)	0.511
Contrast medium (mL)	207.7 (±88.23)	190.25 (±78.64)	155.04 (±70.79)	0.002
Residual SYNTAX score (rSS)	3.69 (±5.73)	6.67 (±7.36)	8.64 (±10.01)	<0.001
rSS > 8	113 (16.3%)	36 (33.3%)	13 (46.4%)	<0.001
SYNTAX index revascularization (%)	80.34 (±26.32)	68.9 (±28.52)	68.99 (±28.31)	<0.001
SRI < 70%	198 (28.4%)	54 (50%)	14 (50%)	<0.001
Staged PCI	173 (24.9%)	31 (29%)	6 (21.4%)	0.593

Data are presented as mean ± SD or n (%). eGFR is expressed in mL/min/1.73 m^2^. eGFR: estimated Glomerular Filtration rate, LAD: Left anterior descending artery coronary, PCI: Percutaneous coronary intervention, rSS: residual SYNTAX score, SRI: SYNTAX index revascularization, SS: SYNTAX score.

**Table 4 jcm-08-00810-t004:** Outcomes at one-year follow-up according to baseline renal function.

Events	eGFR ≥ 60 *n* = 695	eGFR 30-59 *n* = 108	eGFR < 30 *n* = 28	*p* Value
MACE	83 (12%)	28 (25.9%)	10 (35.7%)	<0.001
All cause mortality	25 (3.6%)	12 (11.1%)	7 (25%)	<0.001
Cardiac death	11 (1.6%)	7 (6.5%)	3 (10.7%)	<0.001
STEMI	7 (1%)	1 (0.9%)	0 (0%)	0.866
NSTEMI	33 (4.8%)	8 (7.4%)	2 (7.1%)	0.457
Stroke	9 (1.3%)	6 (4.6%)	0 (0%)	0.036
Stent thrombosis	12 (1.7%)	1 (0.9%)	0 (0%)	0.657
Hemorrhagic events	56 (8.3%)	13 (12%)	5 (17.9%)	0.096
Heart Failure	22 (3.2%)	17 (15.7%)	6 (21.4%)	<0.001
Angina	52 (7.5%)	9 (8.3%)	3 (10.7%)	0.792

Data are expressed as n (%). eGFR is expressed in mL/min/1.73 m^2^. eGFR: estimated Glomerular Filtration rate, MACE: Major adverse cardiac events, NSTEMI: Non ST-segment elevation myocardial infarction, STEMI: ST-segment elevation myocardial infarction.

**Table 5 jcm-08-00810-t005:** Outcomes at one year follow-up according to rSS.

Events	rSS ≤ 8 *n* = 669 (80.5%)	rSS > 8 *n* = 162 (19.5%)	*p* Value
MACE	79 (11.8%)	42 (25.9%)	<0.001
All cause mortality	23 (3.4%)	21 (13%)	<0.001
Cardiac death	8 (1.2%)	13 (8%)	<0.001
STEMI	4 (0.6%)	4 (2.5%)	0.029
NSTEMI	35 (5.2%)	8 (5%)	0.893
CABG	3 (0.4%)	5 (3.1%)	0.002
Stroke	6 (0.9%)	9 (5.6%)	<0.001
Stent thrombosis	11 (1.6%)	2 (1.2%)	0.703
Angina	41 (6.1%)	23 (14.2%)	0.001
Heart Failure	24 (3.6%)	21 (13%)	<0.001
Hemorrhagic events	55 (8.2%)	19 (11.7%)	0.160

eGFR is expressed in mL/min/1.73 m^2^. Data are expressed as *n* (%). eGFR: estimated Glomerular Filtration rate, CABG: Cardiac artery bypass grafting, MACE: Major adverse cardiac events, NSTEMI: Non ST-segment elevation myocardial infarction, STEMI: ST-segment elevation myocardial infarction.

**Table 6 jcm-08-00810-t006:** Univariate and multivariate analysis of risk factors associated with rSS > 8.

	Univariate		Multivariate	
Variables	HR (95% CI)	*p* Value	HR (95% CI)	*P* Value
Age	1.04 (1.03–1.06)	<0.001	1.03 (1.01–1.04)	<0.001
STEMI	0.95 (0.67–1.34)	0.765		
NSTEMI	0.82 (0.73–1.48)	0.819		
Diabetes mellitus	2.14 (1.48–3.11)	<0.001	1.50 (0.97–2.30)	0.064
Hypertension	2.09 (1.43–3.05)	<0.001	1.05 (0.66–1.66)	0.823
Dyslipidemia	1.58 (1.12–2.25)	0.009	0.92 (0.61–1.14)	0.732
Current Smoking	0.63 (0.44–0.91)	0.014		
Obesity	0.93 (0.63–1.38)	0.732		
Atrial Fibrillation	2.01 (1.15–3.53)	0.014	0.99 (0.52–1.92)	0.978
PAD	2.01 (1.14–3.49)	0.015	1.00 (0.52–1.92)	0.983
Prior STEMI	1.62 (0.98–2.67)	0.062		
Prior NSTEMI	2.35 (1.37–4.02)	0.002	1.31 (0.70–2.45)	0.383
Previous angioplasty	1.21 (0.77–1.89)	0.41		
CKD (eGFR < 60 mL/min/m^2^)	3.10 (2.07–4.67)	<0.001	1.75 (1.07–2.86)	0.024
Three-vessel disease	5.05 (3.49–7.33)	<0.001	4.29 (2.89–6.37)	<0.001

eGFR is expressed in mL/min/1.73 m^2^. HR: Hazard ratio, CI: Confidence interval, CKD: Chronic kidney disease, eGFR: estimated Glomerular Filtration rate, PAD: peripheral artery disease, NSTEMI: Non ST-segment elevation myocardial infarction, STEMI: ST-segment elevation myocardial infarction.

**Table 7 jcm-08-00810-t007:** Univariate and multivariate analysis of risk factors associated with cardiac death.

	Univariate		Multivariate	
Events	HR (95% CI)	*p* Value	HR (95% CI)	*p* Value
Age	1.09 (1.05–1.14)	<0.001	1.04 (0.99–1.09)	0.064
Gender (female)	1.27 (0.49–3.27)	0.620		
STEMI	1.07 (0.45–2.56)	0.866		
NSTEMI	1.16 (0.49–2.76)	0.733		
Diabete mellitus	1.29 (0.49–3.31)	0.602		
Hypertension	2.21 (0.81–6.02)	0.123		
Dyslipidemia	1.45 (0.61–3.45)	0.390		
Current Smoking	0.26 (0.07–0.89)	0.031		
Family history of CAD	0.64 (0.19–2.17)	0.474		
Obesity	0.91 (0.33–2.45)	0.844		
Atrial Fibrillation	5.39 (2.07–14.03)	0.001	1.69 (0.59–4.82)	0.326
PAD	4.72 (1.83–12.18)	0.001	1.84 (0.63–5.42)	0.267
CKD (eGFR<60 mL/min)	4.21 (1.77–9.99)	0.001	0.77 (0.28–2.15)	0.627
SBP	0.99 (0.97–1.01)	0.566		
DBP	1.02 (0.98–1.05)	0.380		
Killip ≥ 2	7.02 (2.91–16.95)	<0.001	1.92 (0.67–5.55)	0.226
LVEF ≤ 40%	5.65 (2.38–13.41)	<0.001	1.60 (0.57–4.52)	0.372
Acute Kidney Injury	7.81 (3.31–18.39)	<0.001	2.57 (0.94–6.97)	0.064
Aspirin	0.004 (0.002–0.01)	<0.001		
Colpidogrel	0.87 (0.36–2.11)	0.762		
ACE/ARB	0.08 (0.034–0.191)	<0.001		
Betablocker	0.08 (0.03–0.181)	<0.001		
Statin	0.03 (0.01–0.08)	<0.001		
One-vessel disease	0.26 (0.08–0.76)	0.014		
Three-vessel disease	2.86 (1.20–6.78)	0.017		
Left main	1.06 (0.14–7.89)	0.956		
LAD	2.69 (0.91–7.98)	0.075		
SYNTAX score > 22	4.29 (1.81–10.17)	0.001		
SYNTAX score > 32	6.39 (2.58–15.86)	<0.001		
SRI < 70%	3.55 (1.47–8.56)	0.005		
rSS > 8	7.22 (2.99–17.44)	<0.001	3.38 (1.28–8.93)	0.014

eGFR is expressed in mL/min/1.73 m^2^. CAD: Coronary artery disease, CI: Confidence interval, DBP: Diastolic blood pressure, eGFR: estimated Glomerular Filtration rate, HR: Hazard ratio, LAD: Left anterior descending artery coronary, LVEF: Left ventricle ejection fraction, PAD: Peripheral artery disease, NSTEMI: Non ST-segment elevation myocardial infarction, rSS: Residual SYNTAX score, SBP: Systolic blood pressure, SRI: SYNTAX index revascularization, STEMI: ST-segment elevation myocardial infarction.

**Table 8 jcm-08-00810-t008:** Univariate and multivariate analysis of risk factors associated with MACE.

	Univariate		Multivariate	
Variables	HR (95% CI)	*p* Value	HR (95% CI)	*p* Value
Age	1.03 (1.01–1.04)	<0.001	1.01(0.99–1.02)	0.669
Gender (female)	1.19 (0.80–1.78)	0.386		
STEMI	0.59 (0.41–0.85)	0.005		
NSTEMI	1.84 (1.29–2.63)	0.001	1.74 (1.20–2.53)	0.003
Diabete mellitus	1.68 (1.15–2.44)	0.007	1.10 (0.73–1.67)	0.645
Hypertension	1.96 (1.31–2.94)	0.001	1.33 (0.85–2.09)	0.401
Dyslipidemia	1.37 (0.95–1.96)	0.087		
Current Smoking	0.74 (0.50–1.08)	0.116		
Family history of CAD	0.99 (0.64–1.54)	0.976		
Obesity	0.95 (0.63–1.43)	0.817		
Prior NTSEMI	2.23 (1.37–3.64)	0.001		
Prior STEMI	1.75 (1.08–2.83)	0.022		
Prior unstable angina	1.84 (0.96–3.51)	0.031		
Prior Stroke	1.92 (1.08–3.41)	0.026		
Atrial Fibrillation	2.15 (1.28–3.59)	0.003	1.17 (0.68–2.04)	0.574
PAD	2.13 (1.28–3.51)	0.003	1.11 (0.64–1.92)	0.710
CKD (eGFR<60 mL/min)	2.36 (1.59–3.49)	<0.001	1.26 (0.78–2.02)	0.340
Killip ≥ 2	2.39 (1.65–3.48)	<0.001	1.24 (0.78–1.98)	0.358
LVEF ≤ 40%	2.42 (1.67–3.51)	<0.001	1.74 (1.11–2.71)	0.015
Acute Kidney Injury	2.44 (1.61–3.70)	<0.001	1.46 (0.91–2.35)	0.115
Aspirin	0.03 (0.01–0.05)	<0.001		
Colpidogrel	1.52 (1.06–2.17)	0.021		
ACE/ARB	0.55 (0.38–0.82)	0.003		
Betablocker	0.38 (0.24–0.62)	<0.001		
Statin	0.19 (0.12–0.32)	<0.001		
One-vessel disease	0.49 (0.33–0.71)	<0.001		
Three-vessel disease	2.38 (1.64–3.44)	<0.001		
Left main	2.66 (1.49–4.73)	0.001		
LAD	1.95 (1.29–2.95)	0.002		
SYNTAX score > 22	2.18 (1.51–3.13)	<0.001		
Syntax score > 32	2.67 (1.65–4.32)	<0.001		
SRI < 70%	1.46 (1.01–2.11)	0.040		
rSS > 8	2.46 (1.69–3.58)	<0.001	1.63 (1.08–2.46)	0.02

eGFR is expressed in mL/min/1.73 m^2^. ACE: Angiotensin converting-enzyme inhibitor, ARB: Angiotensin II receptor blockers, CAD: Coronary artery disease, CI: Confidence interval, CKD: Chronic kidney disease, eGFR: estimated Glomerular Filtration rate, HR: Hazard ratio, LAD: Left anterior descending artery coronary, LVEF: Left ventricle ejection fraction, PAD: peripheral artery disease, NSTEMI: Non ST-segment elevation myocardial infarction, rSS: residual SYNTAX score, SRI: SYNTAX index revascularization, STEMI: ST-segment elevation myocardial infarction.

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
