# Peer review of "Prognostic Value of Incomplete Revascularization after Percutaneous Coronary Intervention Following Acute Coronary Syndrome: Focus on CKD Patients"

_jcm, 2019, doi:10.3390/jcm8060810_

Round 1
Reviewer 1 Report
Abstract: please mention number of study population and the 3 groups proportions.
Please mention wether data collection was done in a prospective or retrospective manner.
was kidney function (GFR) also assessed after PCI? were there patients who suffered acute kidney injury due to procedure related contrast agent consumption
Worth notice: very low proportion of patients in the severe CKD group (only 3%!). Patients with normal kidney function are the majority 83.6%
table 1. Group 1 (n=695). But the sum of STEMI (390), NSTEMI (268), and UAP (35) patients is 693 (two patients less). Please check the statistics (consequential error?)
Table 3: (iode volume): use of term "contrast medium" more appropriate
Worth noticing in regard to residual untreated CAD: 46% of the study population had a 1-vessel CAD, which was treated in the primary procedure.
Discussion: Line 231: moderate to high risk ACS: pre-selected population or consecutive ACS? for example IAP with "type A" coronary lesion would not be considered as moderate to high risk PCI/ACS.
Author Response
Abstract: please mention number of study population and the 3 groups proportions.
Changes have been made in the manuscript according to the Reviewer’s concern.
« Methods: 831 ACS patients undergoing PCI were divided into 3 subgroups according to their estimated Glomerular Filtration Rate (eGFR): 695 with eGFR ≥ 60 ml/min/1.73m², 108 with eGFR 60-30 ml/min/1.73m², 28 with eGFR < 30 ml/min/1.73m². »
Please mention whether data collection was done in a prospective or retrospective manner.
A prospective date collection was conducted in this study. To take into account the Reviewer’s comment, this a now more clearly stated in the revised version of the manuscript.
Change: introduction section “In the present prospective study, we sought to determine…”
Was kidney function (GFR) also assessed after PCI? Were there patients who suffered acute kidney injury due to procedure related contrast agent consumption
In the present study, kidney function was assessed after PCI for all patients. However, the specific impact of contrast dye on acute kidney injury was not specifically investigated.
Worth notice: very low proportion of patients in the severe CKD group (only 3%!). Patients with normal kidney function are the majority 83.6%
We thank the Reviewer for his comment acknowledging an important limitation of our study. In the present work, the proportion of patients with severe impaired kidney function appears to be very low. Among others, one possible explanation for this trend relies on the fact that severe CKD patients often had prior coronary artery bypass graft (CABG). As the residual Syntax Score has not been validated for these CABG patients, they were excluded from further analysis in our study. Similarly, the limited number of severe CKD patients is observed in almost all registries. For instance, in the large ACTION registry with 40074 NSTEMI patients treated with PCI, stage 4 and 5 CKD patients only represented 4.6% of the global cohort (1).
Table 1. Group 1 (n=695). But the sum of STEMI (390), NSTEMI (268), and UAP (35) patients is 693 (two patients less). Please check the statistics (consequential error?)
We are sorry for the typo occuring during the preparation of the manuscript, which did not affect the statistics. Indeed as revealed by the Reviewer, two patients are missing in table 1 but were already included in the statistical analysis in the STEMI group. Thie present analysis concerned the 831 patients of the study.
Table 3: (iode volume): use of term "contrast medium" more appropriate
Change has been made according to the Reviewer’s concern.
Worth noticing in regard to residual untreated CAD: 46% of the study population had a 1-vessel CAD, which was treated in the primary procedure.
As pointed out by the Reviewer, we fully recognize that 46% of the patients had a single vessel disease. However, the recruitment of the study is based on a real-life registry including unselected patients. In the large ACTION registry enrolling 12045 CKD patients, one-vessel disease was evidenced in 40.5% of the patients with GFR >60/ml/1.73m2, 30.1% in CKD stage 3 and 23.4% in CKD stage 4 (1).
Discussion: Line 231: moderate to high risk ACS: pre-selected population or consecutive ACS? for example IAP with "type A" coronary lesion would not be considered as moderate to high risk PCI/ACS.
The present study was an observational prospective single-centre study. We included consecutive and unselected patients who were admitted to the coronary angiography unit. No restriction regarding the risk assessment or risk ratings of ACS was made in the patients’ selection of this study. To accommodate the Reviewer, note and given the absence of pre-selected population, the quote “moderate to high risk ACS” has been deleted.
Change. Discussion part. « The current report drawn from a cohort of 831 consecutive ACS patients is the first study … ».
1. Hanna, E.B.; Chen, A.Y.; Roe, M.T.; Wiviott, S.D.; Fox, C.S.; Saucedo, J.F. Characteristics and in-hospital outcomes of patients with non-ST-segment elevation myocardial infarction and chronic kidney disease undergoing percutaneous coronary intervention. JACC Cardiovasc Interv. 2011 Sep;4(9):1002–8.
Reviewer 2 Report
JCM-500783: Evaluation of «Prognostic value of incomplete revascularization after percutaneous coronary intervention following acute coronary syndrome: focus on CKD patients» submitted by Thomas Cardifor et al for publication in JCM.
The aim of this paper was to determine whether chronic kidney disease (CKD) patients exhibit more incomplete revascularization following stage revascularization procedures by PCI.
Authors conclude that in acute coronary syndrome (ACS), CKD is predictive of incomplete revascularization which is a strong predictor of adverse cardiovascular outcomes including cardiac death and major coronary events (MACE).
Major Comments for the authors
1. This is a prospective study on an interesting issue that included 831 CKD patients that underwent PCI and were followed for one year (why not to report this number in the abstract?).
2. The text, the tables and the figures are appropriate and informative about the issue at hand.
3. The references are up to date.
4. Are there any practical implications of the study results? Should we do anything more in those with residual SYNTAX Score (rSS) and eGFR<30 than="" in="" those="" with="" the="" same="" situation="" and="" egfr="">60 or 30-60 ml/min/1.73m²? Should we have a titer follow-up? Should we choose another statin? Should we repeat the PCI? Please comment.
Author Response
1. This is a prospective study on an interesting issue that included 831 CKD patients that underwent PCI and were followed for one year (why not to report this number in the abstract?).
Change has been made according to the Reviewer’s comment.
Change. Abstract. « We evaluated the impact of incomplete revascularization on the occurrence of major cardiovascular events (MACE) at one-year follow-up »
2. The text, the tables and the figures are appropriate and informative about the issue at hand.
3. The references are up to date.
4. Are there any practical implications of the study results? Should we do anything more in those with residual SYNTAX Score (rSS) and eGFR<30 than="" in="" those="" with="" the="" same="" situation="" and="" egfr="">60 or 30-60 ml/min/1.73m²? Should we have a tighter follow-up? Should we choose another statin? Should we repeat the PCI? Please comment.
We would like to thank the Reviewer for his/her positive appreciation.
In patients with significant residual SYNTAX Score (rSS) and eGFR<30 ml/min/1.73m², strategies aimed to achieve the most complete revascularization are recommended. In line with this view and with regards to concomitant co-morbidities that could eventually hamper CABG, CABG or hybrid strategies such as PCI plus CABG or redo PCI should attentively be examined by a dedicate heart team.
In addition, a tighter follow-up is fully recommended and the detection of significant residual myocardial ischemia should be extensively considered.
In this study, most of the patients were treated with high-dose atorvastatin (80 mg). The impact of statin therapy is of importance but beyond the scope of the present study and deserves dedicated investigations.
Reviewer 3 Report
The study is lacking significant impacts or uniqueness over previous reporting studies in their study design and results and does not seem to add significantly more insights to the readers of our journal.
Line 65 - Please define all abbreviations in figure 1.
Line 69 - Please add citation on MDRD formula.
Line 70 - The reason why patients have been divided into three subgroups is unclear. Additionally, it seems that the same patients are mixed in two groups (≥ 60 and 30-60).
Line 78 - It is unclear whether the SYNTAX scores (SS) were assessed at the same time as eGFR was calculated. eGRF was estimated at hospital admission and SS was calculated pre-procedurally. Was it the same time?
Lines 78-91 - I found this description hard to follow so perhaps the authors should rephrase it.
Line 131-133 - I think that a serious limitation of the study is to compare groups with so different numbers.
The fundamental findings of the study, if any, are lost in the mix. The large number of comparisons and analyses ultimately leads to a disjointed analysis.
Author Response
The study is lacking significant impacts or uniqueness over previous reporting studies in their study design and results and does not seem to add significantly more insights to the readers of our journal.
Line 65 - Please define all abbreviations in figure 1.
Abbreviations were defined according to the Reviewer’s concern.
Line 69 - Please add citation on MDRD formula.
A proper citation has been added as requested by the Reviewer.
Change . «Baseline serum creatinine levels were assessed at admission in all patients. The estimated glomerular filtration rate (eGFR) was calculated using the MDRD (Modification of Diet in Renal Disease) formula [6].»
[6] Levey, AS; Bosch, JP; Lewis, JB; Greene, T; Rogers, N; Roth, D. A more accurate method to estimate glomerular filtration rate from serum creatinine: a new prediction equation. Modification of Diet in Renal Disease Study Group. Ann Intern Med. 1999, 461-470. doi: 10.7326/0003-4819-130-6-199903160-00002
Line 70 - The reason why patients have been divided into three subgroups is unclear. Additionally, it seems that the same patients are mixed in two groups (≥ 60 and 30-60).
This type of subdivision is quite usual when addressing the impact of CKD. For instance, in the large CathPCI registry including 283,395 patients, patients were divided into normal renal function (≥ 60), mild CKD (45-59), moderate CKD (30-44) and severe CKD (<30) (1).
In the present analysis, mild and moderate CKD were mixed in one sub-group. This classification is consistent with clinical practice where the risk appears to be maximal in severe CKD patients.
The classification used in the present study is given below:
1. No renal failure: eGFR levels: ≥ 60 ml/min/1.73 m2
2. Mild-Moderate renal failure: eGFR levels ≥ 30 and < 60 ml/min/1.73 m2
3. Severe renal failure: eGFR levels < 30 ml/min/1.73 m2.
Line 78 - It is unclear whether the SYNTAX scores (SS) were assessed at the same time as eGFR was calculated. eGRF was estimated at hospital admission and SS was calculated pre-procedurally. Was it the same time?
We thank the Reviewer for his/her comment. Syntax Score and eGFR were assessed at the same time. We reported the last eGFR before PCI. PCI were systematically performed less than 48 hours after admission.
Lines 78-91 - I found this description hard to follow so perhaps the authors should rephrase it.
The Reviewer’s consideration was taken into account and the whole paragraph was rephrased.
Change : “The SYNTAX Score (SS) was calculated from the pre-procedural angiogram, in which each coronary lesion producing >50% diameter stenosis in vessels >1.5 mm by visual estimation was scored separately using the SS algorithm and added to obtain the overall SS. Staged PCI was performed in patients with angiographic stenosis ≥ 70% or demonstrated residual ischemia assessed either by fractional flow reserve (FFR) or by perfusion myocardial tomography. Staged PCI was performed within 30 days following the index ACS. The residual SYNTAX scores (rSS) was defined as the SS recalculated after staged PCI and was calculated in all patients enrolled in this study. To calculate the rSS, the final post-PCI angiogram was scored to assess untreated disease after staged PCI. A dedicated interventional cardiologist who was blinded to both baseline characteristics and clinical outcomes reviewed all post-procedural angiograms. Similarly, the Syntax revascularization index (SRI) which stands as an angiographic index tool aimed to quantify the proportion of revascularized myocardium was calculated and defined as: 100 (1 - rSS/baseline SS) (%)“
Line 131-133 - I think that a serious limitation of the study is to compare groups with so different numbers.
We agree with the concern of the Reviewer. However, this limitation is a common limitation for almost all the studies focused on CKD. For instance, in the large ACTION registry which enrolled 40074 NSTEMI patients treated with PCI, stage 4-5 CKD patients represented only 4.6% of the global cohort (2). The present data represents a “real world” study with unselected consecutive patients. Therefore, we believe that our data are representative of a normal recruitment in interventional cardiology cathlabs.
The fundamental findings of the study, if any, are lost in the mix. The large number of comparisons and analyses ultimately leads to a disjointed analysis. For these reasons, among others, I believe that this paper is not suitable for its publication in Journal of Clinical Medicine.
The current report drawn from a cohort of 831 consecutive ACS patients is the first study to quantify the extent and severity of coronary artery disease prior and after staged revascularization by PCI according to the extent of kidney disease.
The main finding of this study is that CKD is predictive of incomplete revascularization during ACS as assessed by higher rSS values. The importance of rSS on adverse outcomes is emphasized by the demonstration that incomplete revascularization but not CKD nor diabetes mellitus is a strong independent predictor of 1-year mortality, cardiac mortality and MACE following PCI. The present data point out that incomplete revascularization plays a major role in the determination of adverse outcomes and that CKD and diabetes mellitus are no longer independently associated with cardiac mortality and MACE when rSS is taken into account. The most common accepted reasons for such incomplete revascularization in CKD patients are likely to be multifactorial and may include the following (i) ignorance of NSTE-ACS risk (despite paradoxically high calculated GRACE-risk scores), (ii) ignorance of potential treatment benefit in these patients, (iii) fear of an immediate complication such as contrast induced nephropathy or bleeding, (iv) co-morbidities and (v) lack of definitive population-specific clinical trial data. Altogether we believe that these findings are of clinical interest.
1. Tsai, T.T.; Messenger, J.C.; Brennan, J.M.; Patel, U.D.; Dai, D.; Piana, R.N.; Anstrom, K.J.; Eisenstein, E.L.; Dokholyan, R.S.; Peterson, E.D.; Douglas, P.S. Safety and efficacy of drug-eluting stents in older patients with chronic kidney disease: a report from the linked CathPCI Registry-CMS claims database. J Am Coll Cardiol. 2011 Oct 25;58(18):1859–69
2. Hanna, E.B.; Chen, A.Y.; Roe, M.T.; Wiviott, S.D.; Fox, C.S.; Saucedo, J.F. Characteristics and in-hospital outcomes of patients with non-ST-segment elevation myocardial infarction and chronic kidney disease undergoing percutaneous coronary intervention. JACC Cardiovasc Interv. 2011 Sep;4(9):1002–8.
Round 2
Reviewer 1 Report
The authors thorough response and modifications are highly appreciated. Yet this study is plagued through the limitations as listed, in addition to the less novel core findings of this study, which can be assumed not to be of a high interest to the journal readership.
Reviewer 3 Report
The authors have satisfactorily responded to all my questions and made the necessary changes to the manuscript.
Now, I have only minor comments:
1.The description of Figure 1 has to be corrected according to lines 92-94.
2. Figure 2 should be presented as a blox pot showing individual values.
3. Lines 92-94 “Patients were divided into 3 subgroups according to their eGFR levels: ≥ 60 ml/min/1.73 m2 ; ≥ 30 and < 60 ml/min/1.73 m2 93 ; and < 30 ml/min/1.73 m2”. The authors have to use the same groups descriptions in the text of the manuscript as well as in tables.
4. The authors used different fonts and their size.
5. Furthermore, there are too many results in the paper, so let's focus on the most important (the rest may be showed in supplementary material).
Author Response
The authors have satisfactorily responded to all my questions and made the necessary changes to the manuscript.
Now, I have only minor comments:
1.The description of Figure 1 has to be corrected according to lines 92-94.
2. Figure 2 should be presented as a blox pot showing individual values.
3. Lines 92-94 “Patients were divided into 3 subgroups according to their eGFR levels: ≥ 60 ml/min/1.73 m2 ; ≥ 30 and < 60 ml/min/1.73 m2 93 ; and < 30 ml/min/1.73 m2”. The authors have to use the same groups descriptions in the text of the manuscript as well as in tables.
4. The authors used different fonts and their size.
5. Furthermore, there are too many results in the paper, so let's focus on the most important (the rest may be showed in supplementary material).
We thank the Reviewer for these comments. All changes have been made in the manuscript according to the Reviewer’s concern.